# Relationships between Some Biodiversity Indicators and Crown Damage of *Pinus sylvestris* L. in Natural Old Growth Pine Forests

**Paweł Przybylski [1,\*], Vasyl Mohytych [1] , Paweł Rutkowski [2] , Anna Tereba [1] , Łukasz Tyburski [3] and Kateryna Fyalkowska [1]**

[1]   Forest Research Institute, Braci Leśnej 3, Sękocin Stary, 05-090 Raszyn, Poland;
     v.mohytych@ibles.waw.pl (V.M.); a.tereba@ibles.waw.pl (A.T.); k.fyalkowska@ibles.waw.pl (K.F.)
[2]   Department of Forest Sites and Ecology, Faculty of Forestry and Wood Technology, Poznań University of Life
     Sciences, Wojska Polskiego 71F, 60-625 Poznań, Poland; redebede@wp.pl
[3]   Kampinoski National Park, Tetmajera 38, 05-080 Izabelin, Poland; ltyburski@kampinoski-pn.gov.pl
\*   Correspondence: p.przybylski@ibles.waw.pl

**Abstract:** Biodiversity at the species and individual levels is one of the fundamental elements characterizing an ecosystem. It is assumed that the greater the level of biodiversity, the more tolerant the environment is to changes in external conditions. In recent years, dynamic climate change has negatively impacted the health of many forest trees across Europe, in particular Scots pine. Tree health is commonly characterized by crown defoliation. The study presented here describes and correlates crown defoliation with biodiversity indicators at the species and individual tree levels. Research was conducted in two national parks in Poland (Kampinoski and Bory Tucholskie). Since stands have been under legal protection for many years and forest management is not practiced there, stand development processes taking place there are similar to natural ones. This study provided empirical data on ecosystem response to external stresses based on species and genetic structure. The results confirm differing health of the populations, which results from, among other factors, stand age and the environmental conditions in which they grow. Pine stands in both national parks are genetically diverse but with low genetic variability. Differences in stand health are related to the number of alleles forming the genetic pool. This conclusion is supported by a high correlation coefficient for interactions between defoliation, the number of alleles, and the Shannon index for genotypes. This suggests that greater gene diversity is likely to provide a wider range of phenotypic responses to environmental change.

**Keywords:** biodiversity; SSR markers; ecological risk; defoliation; Scots pine; national park

## 1. Introduction

Biodiversity is defined as the diversity of all living organisms from terrestrial and aquatic ecosystems and the ecological complexes forming part of them within a species, and between species and ecosystems [1] (Convention on Biological Diversity, CBD). The diversity of organisms and their genetic characteristics, together with the abiotic environment, influence each other, creating ecosystems that enable all organisms to live and develop. Maintaining biodiversity in afforestation allows forests to perform ecological, social, and economic functions, and to reduce the risk associated with stand growth.

Genetic diversity of Scots pines (*Pinus sylvestris* L.) depend on the efficiency of crossing processes and allele dispersion in the population [2]. In pines that are naturally genetically variable, where the processes of crossbreeding and seed dispersion are highly effective, it is possible that each individual of the offspring generation will have a different genotype [3]. On the other hand, in pine monocultures formed from an anthropogenically limited number of mother trees, genetic diversity is lowered [4]. For pine, forest management

causes fragmentation of forest sites, reducing biodiversity, gene flow, and, through planting and harvesting, regulates the age of forest stands [5]. Consequently, in the course of forest management works, for example, by planting only progeny plus trees, the selection favors chosen genes (alleles) for particular environments, while excluding other alleles. The adaptation and expression of genes responsible for population survival depends on variable environmental conditions [6]. Thus, in order to understand the role of the main genes, it is necessary to analyze wild populations, preferably in an undisturbed environment [7]. In natural populations that are regenerating without the participation of forest management, we observe a large number of maternal trees (F1), consequently, we do not observe strong selection pressure, and so there is no risk of losing currently necessary genetic information [8], and evolutionary mechanisms spontaneously regulate the gene pool.

An element that is important for forest science is the identification of local adaptation mechanisms [9]. Unfortunately, there are few works related to molecular analyses describing the genetic basis of adaptation [10].

In the present study, analyses were carried out in tree populations (i.e., stands) from two national parks where natural processes have not been disturbed by the influence of the human economy since the parks were established. The species diversity in the area of the Kampinos National Park (KNP), results, in part, from a mosaic of dunes and marshes [11], the park is part of one of the largest inland dune complexes in Poland. Thanks to the proximity of dry and wet areas, a variety of habitats have developed, providing suitable conditions for rare flora and fauna. The area of the Tuchola Forest National Park (Bory Tucholskie National Park, BTNP) is unique due to its habitat values with the presence of numerous lakes of post-glacial origin [12].

Plant biodiversity can be analyzed mainly using a species diversity index, the Shannon index [13]; genetic variability of natural populations can be analyzed using microsatellite nuclear DNA (SSR) markers [14–16]. They are considered to be selectively neutral, but as demonstrated by Wiliams [17], the sequences are located in single-copy regions of DNA, so they can potentially combine with genes subject to negative selection [18]. Consequently, decreased variability of the genetic pool may lead to serious risks for maladaptation [19].

The health condition of stands can be evaluated according to international standards adopted in the ICP Forests and ICP-Focus projects (Programme on Assessment and Monitoring of Air Pollution Effects on Forests) [20], which is assessed on the basis of defoliation of tree crowns. Crown defoliation is also one of the elements used to assess forest health conditions in Poland [21]. The level of defoliation is influenced by the biosocial class of trees, as shown by Dobbertin and Brang [22], and is also dependent on the age of the population [23].

However, there is little work analyzing the health of populations based on defoliation compared with biodiversity at the species and genetic levels. Therefore, the main objective of this research was to compare the impact of species diversity of the forest community and genetic variability of the pine population on forest stand health.

## 2. Materials and Methods

### 2.1. Research Area

Research was conducted in Kampinos National Park (KNP; 52°19′13″ N, 20°47′23″ E) and in Tuchola Forest National Park (BTNP; 53°51′33″ N, 17°31′21″ E) (Figure 1). In each of the national parks, research was carried out in forest stands where the dominant species was Scots pine (*P. sylvestris* L.). The stands were located in areas of strict protection and where forest management was excluded (Table S1 Supplementary Materials). The forest stands were selected taking into account the age and spatial arrangement of the national parks (Figure 1).

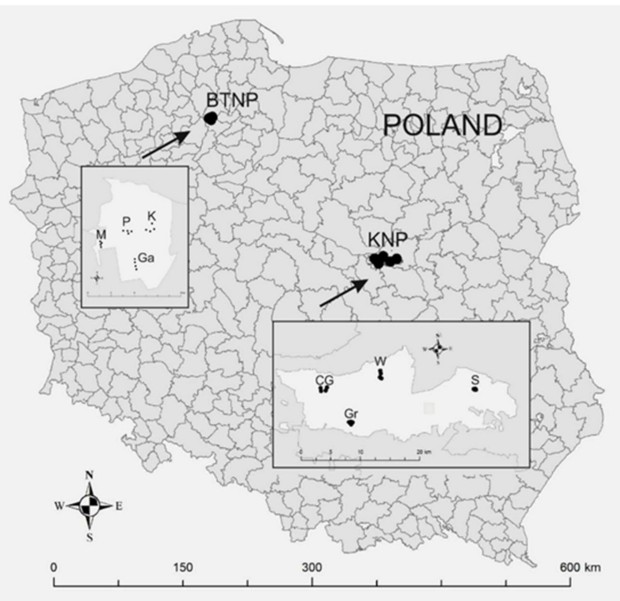

**Figure 1.** Location of national parks and forest areas used to evaluate forest health and species and genetic diversity (50 trees were analyzed in each forest area). KNP—Kampinos National Park (forest areas: CG—Czerwińskie Góry; Gr—Granica; W—Wilków; S—Sieraków); BTNP—Bory Tucholskie National Park (forest areas: M—Mielnica; P—Płęsno; Ga—Gacno; K—Kocioł).

The research areas listed in Section 2.1.1 and Section 2.1.2 were selected to achieve the study objectives.

2.1.1. Kampinos National Park

- Czerwińskie Góry (CG)—species dominating the forest areas: pine about 200–210 years old \*, habitat type: fresh mixed coniferous forest.
- Wilków (W)—species dominating the forest areas: pine about 180–200 years old \*, habitat type: fresh coniferous forest.
- Granica (Gr)—species dominating the forest areas: pine about 160–170 years old \*, habitat type: fresh mixed coniferous forest.
- Sieraków (S)—species dominating the forest areas: pine about 190–200 years old \*, habitat type: fresh coniferous forest.

\* age according to documentation of the Kampinoski National Park (unpublished).

2.1.2. Bory Tucholskie National Park

- Gacno (Ga)—species dominating the forest areas: pine about 140–150 years old \*, habitat type: fresh coniferous forest.
- Płęsno (P)—species dominating the forest areas: pine about 120–130 years old \*, habitat type: fresh coniferous forest.
- Mielnica (M)—species dominating the forest areas: pine about 150–190 years old \*, habitat type: fresh mixed coniferous and fresh coniferous forest.
- Kocioł (K)—species dominating the forest areas: pine about 100–110 years old \*, habitat type: fresh coniferous forest.

\* age according to documentation of the Bory Tucholskie National Park (unpublished).

*2.2. Collection of Plant Material and Observations of Defoliation Level*

In each analyzed forest stand, 50 randomly distributed trees were selected for sampling. If possible, trees were chosen along transects covering the entire forest area. Trees were marked in the field and their geographical coordinates determined with a GPS receiver (Garmin GPSMap 64st). Needles obtained from the selected sample trees were sealed

in Eppendorf-type containers and transported at 4 °C to the laboratory, where they were stored at −20 °C until the time of DNA extraction.

The level of damage to sample tree crowns was assessed in August 2019 based on the number of annual needle classes and the degree of crown defoliation. Defoliation and number of annual needle classes was assessed from the ground simultaneously by two appraisers from opposite sides of the crown with the use of binoculars. Defoliation was calculated as the arithmetic mean of the two evaluators' crown defoliation scores measured on a scale from 1% (no damage) to 100% (dead tree). The methodology is in accordance with international standards adopted in the ICP Forests and ICP-Focus projects [20].

### 2.3. Molecular Analyses

In Kampinos National Park, needles were collected in March 2018, while in Tuchola Forest National Park, they were collected in March 2019. Total genomic DNA was isolated from the collected material using a commercial kit (Macherey-Nagel, Düren, Germany). The quality of the DNA isolate was controlled using 2% agarose gel and a Quawell spectrophotometer. All samples were diluted to 20–30 ng/uL using deionized water and then stored at −20 °C.

Molecular analyses were performed at five polymorphic microsatellite markers [24,25]. The sequences of primers used are shown in Table 1. The forward primers are marked with a set of fluorochromes: VIC, PET, NED, 6-FAM, as shown in Table 1. Five microsatellite loci were amplified in one multiplex reaction. For PCR, we used 1 μL of extracted DNA, 0.2 μL of each primer (10 μM concentration), 5 μL Multiplex PCR Kit (Qiagen), and 2 μL of PCR water. The PCR thermal profile was as follows: 95 °C for 15 min; followed by 35 cycles at 94 °C for 30 s, 60 °C for 90 s, and 72 °C for 90 s; and ending with 60 °C for 30 min. Genotyping was performed using an ABI 3500 Genetic Analyzer (Applied Biosystems, Foster City, CA, USA), and allele lengths were scored using GeneMapper® ver. 5 (Thermo Fisher Scientific Inc., Carlsbad, CA, USA).

**Table 1.** Characteristics of amplified loci of microsatellite nuclear DNA.

| *Loci* | Repeat Motif. | Starter Sequences | Product Size |
|---|---|---|---|
| SPAG 7.14(VIC) | (TG)17(AG)21 | F: TTCGTAGGACTAAAAATGTGTG<br>R: CAAAGTGGATTTTGACCG | 209 |
| SPAC 11.6(NED) | (CA)29(TA)7 | F: CTTCACAGGACTGATGTTCA<br>R: TTACAGCGGTTGGTAAATG | 165 |
| NZPR 11.4(6-FAM) | (CA)15(CA)13(TA)22 | F: AAGATGACCCACATGAAGTTTGG<br>R: GGAGCTTTATAACATATCTCGATGC | 193 |
| SsrPt_ctg4363(VIC) | (AT)10 | F: TAATAATTCAAGCCACCCCG<br>R: AGCAGGCTAATAACAACACGC | 100 |
| PtTX3107(PET) | (CAT)14 | F: AAACAAGCCCACATCGTCAATC<br>R: TCCCCTGGATCTGAGGA | 160 |

### 2.4. Phytosociological Evaluation

Phytosociological assessments were carried out in eight. forest stands (Table S1 Supplementary Materials), using five phytosociological relevès at KNP stands and four phytosociological relevès at BTNP stands. The nomenclature of plant associations follows Matuszkiewicz [26]. Names of taxa of vascular plants were provided according to GBIF [27]. The difference in the number of photos taken at the KNP and BTNP results from the different sizes of the parks. For each stand, the Shannon–Wiener diversity index (*H*) was calculated according to the formula:

$$H = -\sum_{i=1}^{S} p_i \ln p_i \qquad (1)$$

where $p_i = \frac{n_i}{N_i}$. Values of $p_i$ is the ratio of the sum of the degrees of coverage of a given species to the sum of the degrees of coverage of all species; $N_i$ is the sum of the degree of coverage of all species; and $n_i$ is the sum of the degrees of coverage of the *i*-th species.

Each phytosociological relevè was a circular area of 5.64 m radius, resulting in a plot of 100 m². Information for data concerning phytosociological relevè are in Table S2 (Supplementary Materials). The degree of coverage of each species of layers "c" (herbs) and "d" (moss and lichen) within each phytosociological relevè was determined by the Braun–Blanquet method, where relevè plots were assigned a score based on species occurrence, as shown in Table 2. Braun–Blanquet scores were assigned to values shown in Table 2 for calculation of the Shannon–Wiener diversity index. Because all research plots were established in pine stands, the Shannon–Wiener diversity index (*H*) was determined by taking into account only species of herbaceous plants, mosses, and lichens.

**Table 2.** Transformation of Braun–Blanquet cover-abundance scores to percentage cover.

| Braun–Blanquet Score | Value Taken for Calculations |
|:---:|:---:|
| r | 0.1 |
| + | 0.5 |
| 1 | 2.5 |
| 2 | 12.5 |
| 3 | 37.5 |
| 4 | 62.5 |
| 5 | 87.5 |

### 2.5. Statistical Analysis

#### 2.5.1. SSR Analysis

The values of the following attributes were calculated using GenALEx 6.5 [28]: number of alleles (*Na*), effective number of alleles, Shannon's information index (*I*), observed heterozygosity (*Ho*), expected heterozygosity (*He*), Wright's coefficient (*F*), and inbreeding coefficients (*Fis*). The *p* value for each locus and population was calculated for the *Fis* factor value using Fstat ver. 2.9.3 [29]. By using Arlequin ver. 3.5 [30], the *Fst* genetic distance value, including the *p* value for this factor, were calculated between tested populations. The principal coordinates analysis (PCoA) in the GenALEx 6.5 program was carried out based on the value of *Fst*, the fixation index providing an estimation of genetic differentiation of the tested populations [28].

#### 2.5.2. Phytosociological Analysis

In order to examine the differences in plant diversity between the eight forest stands, the Shannon–Wiener diversity index (*H*) was calculated collectively for each plot, summing the values of the degree of stand coverage for each species in phytosociological photos of the plot. The standard deviation was calculated for each stand based on the Shannon–Wiener diversity indicators from the phytosociological relevès of each stand.

The plant species identified in the phytosociological relevès were compared with indicator plant species described in old growth forests in Poland by Dzwonko and Loster [31].

#### 2.5.3. Stand Health Analysis

One-way analysis of variance (ANOVA) was used to compare population (stand) means for stand health traits (defoliation and number of needle year classes). The analyses were carried out in R [32] according to the next model:

$$y_{ij} = P_i + e_{ij},$$

where, $y_{ij}$ is the *j*-th observation of the trait in the *i*-th population, $P_i$ is the mean of the *i*-th population, and $e_{ij}$ is the error of the *j*-th observation in the *i*-th population. Tukey's

post-hoc test was carried out in R using the "laercio" statistical package [33], correlation analysis using "corrplot" [34], and PCA analysis using "ggbiplot" [35].

## 3. Results

### 3.1. Health Condition of Tree Crowns

The assessment of forest health indicated significant differentiation between stands in crown defoliation and the number of annual needle classes of the analyzed trees (Table 3). The results indicate a higher damage level in stands at KNP compared to those at BTNP. The highest level of damage was observed in Granica and the lowest in populations (stands) at Płęsno and Kocioł (Table 4). This is a strong negative correlation ($-0.76$; $p \leq 0.05$), which means that high defoliation goes with low LRIgiel (number of annual needle classes) (Figure 2). The resulting differences in health status between populations justify further stages of comparative analyses.

**Table 3.** Analysis of variance of attributes of population health based on crown defoliation and number of annual needle classes (LRIgiel).

|  |  | Df | Sum. Sq | Mean Sq | *F* Value | Pr (>F) |
|---|---|---|---|---|---|---|
| Defoliation | Population | 7.00 | 10,466.00 | 1495.20 | 11.26 | $5.22 \times 10^{-13}$ *** |
|  | Residuals | 390.00 | 51,794.00 | 132.80 |  |  |
| LRIgiel | Population | 7.00 | 8.78 | 1.25 | 2.96 | 0.00481 ** |
|  | Residuals | 390.00 | 164.77 | 0.42 |  |  |

Significance levels: ** $p \leq 0.01$, *** $p \leq 0.001$. Df—degrees of freedom; Sum Sq—sums of squares; Mean Sq—mean squares; *F* value—test statistic; Pr(>F)—*p*-value.

**Table 4.** Assessment of significant differences in attributes of stand health based on Tukey's post-hoc test. Populations with the same letters do not differ significantly. LRIgiel is the number of annual needle classes.

| Populations | Defoliation (%) | LRIgiel |
|---|---|---|
| Granica | a(46.9) | a(1.52) |
| Cz. Góry | ab(41.1) | ab(1.78) |
| Wilków | abc(39.9) | b(1.92) |
| Sieraków | bc(38.7) | ab(1.86) |
| Gacno | bcd(35.5) | b(2.00) |
| Mielnca | cd(33.3) | b(1.92) |
| Kocioł | d(31.2) | b(2.02) |
| Płęsno | d(31.1) | ab(1.81) |

### 3.2. Biodiversity at the Genetic Level

On average, 15.05 alleles per population were identified. Comparing parks, populations in BTNP had a higher number of alleles, averaging 16.5 alleles per population. In comparison, KNP averaged 13.6 alleles per population (Table 5). The richest allelic diversity are in the populations at Kocioł (18.0 alleles on average) and Gacno (17.2 alleles on average) from BTNP, while the poorest allelic diversity are at Granica (13.2 alleles on average) and Czerwińskie Góry and Wilków stands (13.4 alleles each) from KNP (Table 5). The richness of alleles was converted into values of effective number of alleles (*Ne*) and the Shannon index (*I*), whose values in the populations coincide with the number of alleles (*Na*).

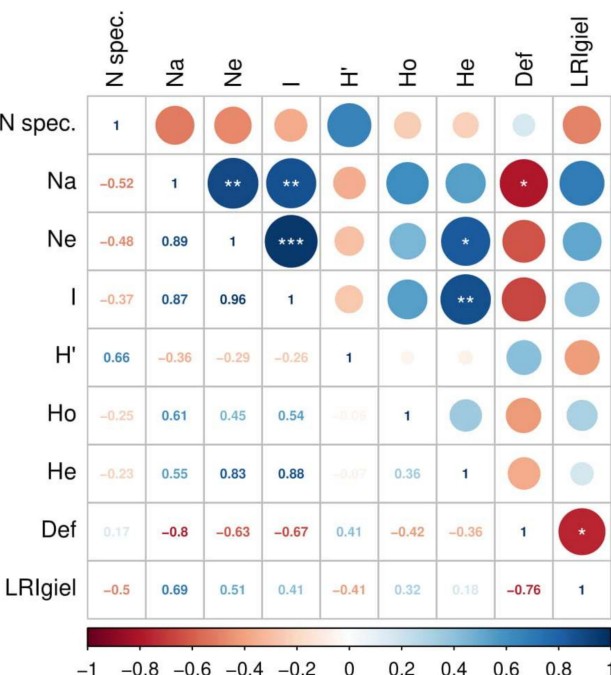

**Figure 2.** Correlations among attributes of biodiversity and stand health for all analyzed populations. The attributes analyzed include: Nspec.—number of species, *Na*—number of alleles, *N*—effective number of alleles, *I*—Shannon index (genotypes), *H′*—Shannon–Wienner index (species), *Ho*—observed heterozygosity, *He*—expected heterozygosity, Def.—crown defoliation coefficient, LRIgiel—number of annual needle classes. Statistical significance levels: * $p \leq 0.05$, ** $p \leq 0.01$, *** $p \leq 0.001$.

**Table 5.** Biodiversity in forest stands at two national parks.

| Population | N Spec. | *Na* | *Ne* | *I* | *Ho* | *Fis* |
|---|---|---|---|---|---|---|
| Sieraków KNP | 25 | 14.400 | 8.438 | 2.202 | 0.579 | 0.324 * |
| S.E. | | 2.731 | 2.156 | 0.256 | 0.080 | |
| Cz. Góry KNP | 37 | 13.400 | 7.778 | 2.151 | 0.764 | 0.101 |
| S.E. | | 2.619 | 1.868 | 0.235 | 0.064 | |
| Granica KNP | 36 | 13.200 | 8.093 | 2.160 | 0.644 | 0.244 * |
| S.E. | | 2.267 | 2.001 | 0.247 | 0.059 | |
| Wilków KNP | 21 | 13.400 | 7.670 | 2.081 | 0.627 | 0.246 * |
| S.E. | | 2.600 | 2.145 | 0.276 | 0.064 | |
| BTNP Kocioł | 15 | 18.000 | 9.864 | 2.384 | 0.779 | 0.110 * |
| S.E. | | 4.528 | 2.706 | 0.274 | 0.044 | |
| BTNP Gacno | 16 | 17.200 | 8.713 | 2.222 | 0.764 | 0.090 * |
| S.E. | | 4.620 | 2.665 | 0.325 | 0.074 | |
| BTNP Płęsno | 33 | 15.800 | 8.375 | 2.242 | 0.723 | 0.145 * |
| S.E. | | 3.292 | 2.415 | 0.278 | 0.026 | |
| BTNP Mielnica | 49 | 15.000 | 8.364 | 2.187 | 0.656 | 0.224 * |
| S.E. | | 3.564 | 2.346 | 0.289 | 0.086 | |
| Total | 29 | 15.050 | 8.412 | 2.204 | 0.692 | 0.185 * |
| S.E. | | 1.117 | 0.745 | 0.089 | 0.024 | |

Nspec.—number of species, *Na*—number of alleles, *Ne*—effective number of alleles, *I*—Shannon index (genotypes), *Ho*—observed heterozygosity, and *Fis*—coefficient of inbreeding. Significance levels: * $p \leq 0.05$.

In all analyzed populations, the *F* factor had a positive value, which means a higher-than-expected homozygous frequency (Table 5). The value of *Fis*, the inbreeding coefficient, was statistically significant for all populations (Table 5) except for Czerwińskie Góry from KNP (Table 6), where the *Fis* coefficient was equal to 0.101. The obtained data may result from the presence of null alleles, the presence of which was demonstrated for three out of four loci. However, their number is similar to that in other studies conducted in pine stands

(Table S3 Supplementary Materials). For the analyzed loci, the highest heterozygosity was in populations from Kocioł (*Ho* = 0.77) and Gacno (*Ho* = 0.76) from BTNP; in comparison, in KNP, the Czerwińskie Góry stand was characterized by equally high genetic variability (*Ho* = 0.76) despite a relatively small number of alleles in the population (*Na* = 13.40) (Table 5). The lowest heterozygosity was observed in the population at Sieraków (*Ho* = 0.57) at KNP.

**Table 6.** Comparison of genetic differentiation (*Fst*) between populations. Statistically significant differences are shown above the diagonal.

| | Sieraków KNP | Cz. Góry KNP | Granica KNP | Wilków KNP | BTNP Kocioł | BTNP Gacno | BTNP Płęsno | BTNP Mielnica |
|---|---|---|---|---|---|---|---|---|
| Sieraków KNP | 0.000 | ns | ns | ns | ns | ns | 0.014 | 0.028 |
| Cz.Góry KNP | 0.010 | 0.000 | ns | ns | ns | 0.044 | ns | 0.024 |
| Granica KNP | 0.010 | 0.007 | 0.000 | ns | ns | ns | 0.014 | ns |
| Wilków KNP | 0.008 | 0.009 | 0.011 | 0.000 | ns | ns | 0.000 | 0.001 |
| BTNP Kocioł | 0.008 | 0.008 | 0.007 | 0.008 | 0.000 | ns | ns | ns |
| BTNP Gacno | 0.006 | 0.009 | 0.008 | 0.006 | 0.007 | 0.000 | 0.004 | 0.003 |
| BTNP Płęsno | 0.010 | 0.009 | 0.012 | 0.016 | 0.012 | 0.010 | 0.000 | ns |
| BTNP Mielnica | 0.011 | 0.011 | 0.009 | 0.014 | 0.010 | 0.011 | 0.006 | 0.000 |

The carried out statistical analysis for *Fst* showed significant differences between 8 out of 28 possible compilation variants (Table 6). In all significant variants, the populations were from different national parks, except for the Gacno and Mielnica stands growing in BTNP.

### 3.3. Biodiversity at the Species Level

In total, in both national parks, phytosociological studies identified 84 species, including vascular plants, bryophytes, and terrestrial lichens. The number of species in each category are shown in Table 7. Differences among populations in the number of species, the number of species representative of old growth stands, and the Shannon–Wiener diversity index (*H′*) are shown in Table 8.

### 3.4. Correlation between Biodiversity and Stand Health

The comparison of forest stands presented in Figure 3 using PCA analysis, taking into account biodiversity and stand health, shows the distinctiveness of the studied forest ecosystems in both parks. One of the main differentiating criteria was the health of stands relative to their age. Diversity at the genetic and species levels was an additional differentiating criterion. Hence, the observed scatter of plots in Figure 3 reflects the synergistic influence of all components.

**Table 7.** Numbers of species of vascular plants, bryophytes, and lichens in BTNP and KNP.

| Number of Species | BTNP | KNP | BTNP + KNP |
|---|---|---|---|
| Total number of species | 59 | 55 | 84 |
| Vascular plant species | 45 | 53 | 69 |
| Bryophytes species | 11 | 2 | 12 |
| Lichen species | 3 | 0 | 3 |
| Shannon–Wiener index (*H′*) | 2.14137 | 2.29675 | 2.40883 |

**Table 8.** A comparison of the number of species.

| National Park | Surface | Number of Species in the Plot | Number of old Growth Forest Species | Age of the Dominant *P. sylvestris* L. in the Stand | H′ | SD |
|---|---|---|---|---|---|---|
| KNP | CG | 36 | 12 | 200–210 (avg. 205) | 1.79144 | 0.31258 |
| | Gr | 34 | 15 | 160–170 (avg. 165) | 1.96834 | 0.28773 |
| | W | 21 | 10 | 180–200 (avg. 190) | 1.36353 | 0.09822 |
| | S | 25 | 10 | 190–200 (avg. 195) | 1.83306 | 0.36663 |
| BTNP | M | 48 | 15 | 150–190 (avg. 170) | 2.10052 | 0.28560 |
| | Ga | 16 | 3 | 140–150 (avg. 145) | 1.68053 | 0.06668 |
| | K | 15 | 3 | 100–110 (avg. 105) | 1.51323 | 0.13578 |
| | P | 33 | 4 | 120–130 (avg. 125) | 1.96201 | 0.37896 |

Shannon–Wiener diversity index (H′), and standard deviations for individual forest stands (CG—Czerwińskie Góry; Gr—Granica; W—Wilków; S—Sieraków; M—Mielnica; P—Płęsno; Ga—Gacno; K—Kocioł).

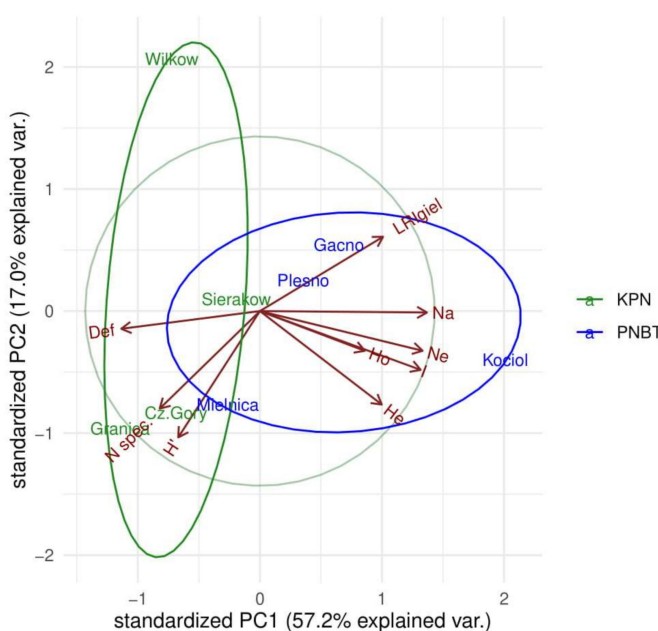

**Figure 3.** Comparison of forest stands using PCA analysis of biodiversity attributes and stand condition.

Correlations between biodiversity and stand health across all populations are presented in Figure 2. There was a significant ($p < 0.05$) and very strong negative correlation (−0.8) between defoliation level and the number of alleles (*Na*), as well as a high correlation (0.69) between the number of annual needle classes present and the number of alleles. Moderately strong negative correlations were found between both genetic variability (*Ho*) and species diversity (H′) with defoliation level.

## 4. Discussion

### 4.1. Defoliation

Crown defoliation is one of the basic criteria for forest monitoring in Europe. The level of defoliation is affected by tree genotype and external factors affecting the plant [36]. The assessment of defoliation is an inexpensive way to determine the degree of stand damage [37], hence the widespread use of this method. However, it should be remembered that crown condition does not fully describe tree condition, as trees may die with undamaged crowns and stands may recover from severe defoliation. For these reasons, when discussing the health of studied populations, a number of factors in addition to crown condition should be analyzed, such as age, habitat, and external stress factors. Stand age plays

an important role in the health of stands [38]. Younger populations, in particular those up to 60 years of age, are significantly less damaged than older stands, which is confirmed by results from the present study. The younger BTNP populations are characterized by lower levels of crown defoliation. However, it should be noted that all of the populations evaluated in this study were older trees, with well-developed root systems [39]. Trees growing in fertile soils, if exposed to a water deficit, may be at significant physiological risk because of root system architecture [40], and Scots pine in fertile soils are usually outcompeted by other forest tree species. Water deficits affect the basic metabolic processes of plant cells [41]. A significant water deficit was observed at KNP in 2015, when annual precipitation was 406 mm [42]. In subsequent years precipitation was higher at about 600 mm. A greatly reduced assimilation apparatus is an additional stress factor for pine, i.e., a reduction of water flux in xylem produced by decreased transpiration in leaves due to stomatal closure.

### 4.2. Species Diversity

Species richness in the present study was evaluated only in pine stands and therefore does not reflect the full range of species diversity present in either national park. For example, 1400 species of vascular plants have been found in KNP [43].

Species diversity of "old growth" forests was described by Dzwonko and Loster [31]. However, those authors developed a list of species for deciduous forests, while no such undertaking has been done for old growth pine forests. In the present study, a strong correlation (0.7) was found between the number of species included as the representative for old forests and the age of the studied stands, wherein KNP had on average more species (and at the same time was older) than stands in BTNP (11.75 compared to 6.25, respectively, for KNP and BTNP, with an average age of 188 years in KNP and 136 years in BTNP).

Some of the plant species associations that were characteristic of old growth pine forests in this study have been described as being phytosociological associations typical of some deciduous forest types. Three species (out of a total of 20), i.e., *Anemone nemorosa* L., *Carex digitata* L., and *Poa nemoralis* L., are characteristic of the *Querco-Fagetea* vegetation classification, one (*Viola reichenbachiana* Jord. ex Boreau) is characteristic of the Fagetalia order and is included in the *Querco-Fagetea* class, two species are characteristic of the *Quercetea robori-petraeae* (*Hieracium sylvaticum* Lam., *Lathyrus linifolius* (Reichard) Bässler) classification, and three are characteristically found in lowland oak forests (*Luzula pilosa* Willd., *Pteridium aquilinum* (L.) Kuhn, and *Lysimachia europaea* (L.) U.Manns & Anderb.) [43]. This is important in interpreting the potential natural stand succession that will take place over time in both parks. Currently, where pine forests are dominant, and taking into consideration current species composition, stand age, and the correlation between stand age and the number of species characteristic of an old growth stand, it can be assumed that a slow succession of pine forests to deciduous forests is taking place, for which the *Querco roboris-Pinetum* type, dominant in the research plots, is a transitional stage. However, the current forest stand type can also be interpreted as a stable system, in which a characteristic feature of the *Querco roboris-Pinetum* unit is a peculiar combination of species from the *Vaccinio-Piceetalia* order, with a component of species with wider ecological amplitude, passing from the *Querco-Fagetea* class, as described by Matuszkiewicz [26].

### 4.3. Genetic Diversity

4.3.1. Genetic Diversity of Populations

The genetic markers used in the present study are probably not located in DNA coding regions, but Williams [17] has shown that SSR markers can be coupled with genes subject to negative selection. Therefore, if some of the markers used in this work are directly related to genes, their reduction in the genetic pool will have adaptive consequences. The results of this study confirm the usefulness of the microsatellite DNA (SSR) markers used due to their polymorphism. In all analyzed populations, the frequency of the majority of alleles was above 3%, as in Robledo-Arnuncio et al. [44]. On the other hand, in each population there

were private alleles with a frequency below 1% (rare alleles). In the current study, stands in BTNP had a richer genetic pool and differences resulted from a higher frequency of rare alleles. Authors such as Raja et al. [45] claim that in forest tree populations, new alleles with low frequency of occurrence are present in the pollen cloud, giving the stands additional adaptation possibilities. This is consistent with results obtained in the present study.

The studied populations were characterized by a lower than expected heterozygosity coefficient than normally found in Poland ($Ho = 0.80$) by Nowakowska [46]. This result is typical for populations with a reduced number of individuals, where inbreeding is more likely. The observed deficiency of heterozygotes is contradictory to the usual observation of excess of heterozygotes in pine stands, although examples of decreased heterozygosity and impoverishment of alleles in managed stands of *Pinus strobus* L. were described by Rajora et al. [15]. The studied populations are among the oldest pine stands in the country. According to historical data (data from KNP, unpublished), stands that are currently 200 years old were formed naturally from a small number of mother trees. The regeneration of the stand of a small number of mother trees is conducive to the development of genetic drift and the past occurrence of the bottle neck effect. In populations such as those analyzed in this article, an excess of homozygotes is expected and probably a result of reproduction between related trees—the Wahlund effect [47]. The second possible reason for the excess of homozygots in the analyzed populations are "null" alleles. Their presence was proven and described for the obtained results (Table S3 Supplementary Materials).

The genetic diversity between the examined populations was insignificant and generally below 1%. This result is typical for pine stands from Europe and Poland [48,49]. In a comprehensive evaluation of pine differentiation in Poland, Nowakowska [46] obtained the highest coefficient *Fst* for a population from the Baltic Land ($Fst = 0.036$) and the lowest for the Silesian Land ($Fst = 0.013$).

4.3.2. Interactions Observed

Preserving the genetic variability of forest tree populations is considered a safety net that allows for adaptation to the projected climate change that threatens forest ecosystems [50]. If plants react too slowly to changes in growth conditions caused by climate change, their ecological stability may be affected [51]. The ability of trees to adapt to changing growth conditions is generally influenced by a small percentage of variability occurring in a population after exposure to a stress factor. Hence, the plasticity of a single plant is limited by the fact that the genetic variability is not unlimited. This means that the population achieves its adaptability through the genetic variability present between individuals [52]. Studies by Rankevich et al. [53] indicate that under stress conditions the heterozygosity of a population may be higher. However, this was not confirmed in the present study, which is in line with Kopp et al. [54], who claim that the relationship between stress tolerance and heterozygosity is different. According to Kopp et al. [54], stress factors increase the frequency of specific adaptive alleles, while the total heterozygosity of the population decreases.

Phenotypic plasticity, which indicates the extent to which environmental variability can modify the expression of individual alleles, can also play a role in adapting to climate change. Using phenotypic plasticity, plants can react to environmental changes without genetic changes. This phenomenon is probably the result of differences in allele expression in different environments [55], as well as changes in interactions between alleles and the environment. Plasticity does not depend on heterozygosity [56]; however, the magnitude of this response is referred to as the genotype–environment interaction. In the present study, a significant interaction between the number of alleles and the amount of crown defoliation was obtained, confirming the ability of plants to respond to stress.

5. Conclusions

The stands in this study are among the most valuable in terms of natural processes in Poland. Additionally, due to law restriction processes of secondary succession, natural

selection and gene flow take place and are unaffected by forest management. This fact allows the populations to be used to observe biological phenomena separate from direct human effects. Climate change observed in recent years has had an impact on the level of damage and stand death, including among pine populations. The dynamics of stand death are observed based on the defoliation of tree crowns. However, biodiversity provides natural protection of ecosystems against stress factors. The present study showed differences in the level of crown damage in the populations, which are attributed mainly to stand age. An additional factor reducing the scale of stand damage was the number of alleles in the studied populations. A larger number of gene forms allows for a wider range of phenotypic responses. In old growth trees with a diverse genetic structure, where selection phenomena favor certain alleles, genetic richness may determine stand stability over time. Moreover, this study showed a high correlation between crown defoliation and both the number of alleles and the Shannon index for genotypes. No significant correlations were found between crown defoliation and the number of plant species or the Shannon–Wienner index of species diversity.

**Supplementary Materials:** The following are available online at https://www.mdpi.com/2071-1050/13/3/1239/s1, Table S1. Geographical coordinates of phytosociological relevès; Table S2. Phytosociological relevés. Name of study plots: M (Mielnica-BT NP.), Ga (Gacno-BT NP.), K (Kocioł-BT NP.), P (Płęsno BT NP.), P (Płęsno-BT NP.), C (Sieraków-K NP.), Gr (Granica-K NP.), W (Wilków-K. NP.), CG (Czerwińskie Góry-K NP.), numbers denote the ordinal number within the a single study plots; Table S3. Null alleles analyses by Micro-Checker v2.2.3. (van Oosterhout i in. 2004).

**Author Contributions:** Conceptualization, P.P.; methodology, P.P., V.M., P.R., and A.T.; software, V.M. and A.T.; writing—original draft preparation, P.P.; writing—review and editing, P.R., A.T., and Ł.T.; visualization, V.M. and K.F.; supervision, P.P.; project administration, P.P. and Ł.T. All authors have read and agreed to the published version of the manuscript.

**Funding:** This research was funded by PGL LP, grant number DE/373-180/2019 (KNP) and grant number 67 02 54 (46/2019) (BTNP).

**Institutional Review Board Statement:** Not applicable.

**Informed Consent Statement:** Not applicable.

**Data Availability Statement:** Data publicly available in annual reports held in the library of the Forest Research Institute in Poland.

**Conflicts of Interest:** The authors declare no conflict of interest. The funders had no role in the design of the study; in the collection, analyses, or interpretation of data; in the writing of the manuscript, or in the decision to publish the results.

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
