# Peer review of "Relationships between Some Biodiversity Indicators and Crown Damage of Pinus sylvestris L. in Natural Old Growth Pine Forests"

_sustainability, doi:10.3390/su13031239_

Round 1

Reviewer 1 Report

The work is interesting and well organized.
Only a few clarifications, also indicated in the attached text together with other minor ones.

Page 5 The phytosociological data table should be reported in the supplementary materials

Page 6. - In table 3 it is advisable to specify the meaning of the acronomics

Page 10. The Latin names of the phytosociological syntaxa go in italics, as well as the names of the species.

The class "Quercetea-petraeae" is not considered by most authors, perhaps you mean "Quercetea robori-petreaea" or "Quercetea pubescenti-petraeae"

Page 10. You should better explain the congruence between the results of this study and the preliminary one (41.
Tyburski, Ł.; Przybylski, P. Health condition of the Scots pine (Pinus sylvestris L.) in Kampinos National Park – preliminary studies. Folia Forestalia Polonica. 2016, 58 (4), 240 – 245).

Author Response

Thank you for the kind words about work. All suggestions related to the improvement of the publication have been included and are found in the text.

Page 5 The phytosociological data table should be reported in the supplementary materials. The  phytosociological data table has been added to supplementary 2. Information about this was added to the text on page 5.  

Page 6. - In table 3 it is advisable to specify the meaning of the acronomics. Table 3 has been corrected to clarify the meaning of acronyms

Page 10. The Latin names of the phytosociological syntaxa go in italics, as well as the names of the species. The spelling forms of phytosociological syntaxa were changed.

The class "Quercetea-petraeae" is not considered by most authors, perhaps you mean "Quercetea robori-petreaea" or "Quercetea pubescenti-petraeae" Done.

Page 10. You should better explain the congruence between the results of this study and the preliminary one (41.Tyburski, Ł.; Przybylski, P. Health condition of the Scots pine (Pinus sylvestris L.) in Kampinos National Park – preliminary studies. Folia Forestalia Polonica. 2016, 58 (4), 240 – 245). Corrections were introduced in the text

Reviewer 2 Report

It is a decent quality research paper dealing with one of the most topical issues, a sustainability of Scots pine stands which represent a substantial portion of boreal forests. The research uses appropriate methods and covers large enough number of study samples to achieve its objectives and arrive to well-based conclusions. Still, there are some points to be considered to improve the manuscript. Below is a line-by-line list of them.

Title: Based on the contents of the work I suggest a more explicit title, for example: "Relationships between some biodiversity indicators and crown damage of Pinus sylvestris in natural old growth pine forests".

Lines 36-37: "Maintaining biodiversity in afforestation allows forests to perform ecological, social and economic functions, and to reduce the risk of growing forest trees" – please clarify the last part of the sentence by rewording it.

Line 38: Replace "Genetic diversity scotch pine" with "Genetic diversity in Scots pine". (Use common name Scots with capital first letter).

Line 51-52: "and evolutionary mechanisms regulate one another or forest sciences, an important element is the identification of local adaptation mechanisms" – please clarify what do you mean with this sentence.

Lines 53, 54: Replace "analyzes" with "analyses".

Line 70: Provide explanation for the abbreviation "ICP".

Line 71: Replace "used asses" with "used to assess".

Lines 85, 146: Replace "Annex 1" with "Table 1S, Supplementary Materials)" if you refer to that supplement.

Lines 82, 86, 251, 255, 258: Replace "Fig." with "Figure".

Line 86: Replace "research areas" with "forest areas".

Line 121: Describe briefly how did you establish the "number of annual needle classes".

Line 158: Replace "5.64 m2 radius" with "5.64 m radius".

Line 106: Replace "yers old" with "age".

Line 109: Replace "stands" with "forest areas".

Lines 146-147: Were the total numbers of phytosociological photos 20 at KNP and 16 at BTNP? It should be indicated in the text too.

Lines 166-167: Correct the title of Table 2 to "Transformation of Braun-Blanquet cover-abundance scores to percentage cover"

Line 174: Replace "Arleqiun" with "Arlequin".

Line 187: Use "Defoliation analysis" as a third level heading instead of "ANOVA".

Line 205 and Table 3: Which is correct "LRIgel" or "LRIgiel"? I would suggest using a meaningful English abbreviation instead.

Line 206: Replace "Significant codes: 0 '***' 0.001 '**' 0.01 '*' 0.05 '.' 0.1 ' ' 1" with "Significance levels: ** – 0.01; *** – 0.001".

Line 208: Replace "similar" with "the same".

Lines 213 and 215: Use "allelic diversity" instead of "allelic stands".

Line 237 and Table 6: Please make sure whether the statement "bolded font above the diagonal" is correct. Also, the last column should be moved to the first position.

Table 7: Shannon-Wiener index (H') should be positive, not negative.

Figure 3: It is a bit overloaded presentation of correlations: the (not necessary) colors are explained by the sidebar color scale, while the sizes of circles are not. I suggest removing colors and circles as a redundant information and making a simple plain text table instead.

Line 270: Replace "=" with "<".

Line 274: "range of defoliation is affected by tree genotype and phenotype" – By "range", do you mean the degree (extent) of defoliation? Also, I would doubt whether defoliation is affected by phenotype.

Lines 274-275: When referring to the "external factors affecting the plant" I would mention atmospheric pollution as one of the major factors affecting P. sylvestris.

Line 281: "Stand age plays an important role in the health of stands in Poland" – Simply, it is just a common rule for biological species and not only in a given country.

Lines 285-286: Replace "in which a well-developed a root system" with "with well-developed root systems".

Lines 286-287: "Trees growing in fertile soils, if exposed to water deficit, may be at significant physiological risk because of root system architecture". Here I would add that Scots pine in fertile soils is usually outcompeted by other forest tree species.

Line 289: "water deficit was observed at KNP" – Do you mean annual precipitation deficit? Then correct, respectively.

Line 374: "Additionally, due to restrictions on the processes occurring in these stands" – What kind of restrictions do you mean here? If it is meant restrictions on human intervention, correct it, respectively.

Line 390: Replace "Table 1" with Table 1S".

Line 391: "Geographical coordinates phytosociological photos" – Do you mean "Geographical coordinates of phytosociological photos"?

Lines 392-405: Remove quotation-marks when providing your own statements.

References: Use boldface fonts for years of each reference.

Author Response

Thank you for the kind words about work. All suggestions related to the improvement of the publication have been included and are found in the text.

Title: Based on the contents of the work I suggest a more explicit title, for example: "Relationships between some biodiversity indicators and crown damage of Pinus sylvestris in natural old growth pine forests". The title has been corrected.

Lines 36-37: "Maintaining biodiversity in afforestation allows forests to perform ecological, social and economic functions, and to reduce the risk of growing forest trees" – please clarify the last part of the sentence by rewording it. The last part of the sentence has been corrected.

Line 38: Replace "Genetic diversity scotch pine" with "Genetic diversity in Scots pine". (Use common name Scots with capital first letter). Done

Line 51-52: "and evolutionary mechanisms regulate one another or forest sciences, an important element is the identification of local adaptation mechanisms" – please clarify what do you mean with this sentence. Thank you for pointing out the lack of logic in the presented sentence. Errors occurred while editing the language of the text. The text has been corrected.

Lines 53, 54: Replace "analyzes" with "analyses". Done

Line 70: Provide explanation for the abbreviation "ICP". Done

Line 71: Replace "used asses" with "used to assess". Done

Lines 85, 146: Replace "Annex 1" with "Table 1S, Supplementary Materials)" if you refer to that supplement. Done

Lines 82, 86, 251, 255, 258: Replace "Fig." with "Figure". Done

Line 86: Replace "research areas" with "forest areas". Done

Line 121: Describe briefly how did you establish the "number of annual needle classes". The information has been supplemented in the text.

Line 158: Replace "5.64 m2 radius" with "5.64 m radius". Done

Line 106: Replace "yers old" with "age". Done

Line 109: Replace "stands" with "forest areas". Done

Lines 146-147: Were the total numbers of phytosociological photos 20 at KNP and 16 at BTNP? It should be indicated in the text too. Information on the number of phytosociological photos is provided in the text in lines 172-174.

Lines 166-167: Correct the title of Table 2 to "Transformation of Braun-Blanquet cover-abundance scores to percentage cover" Done

Line 174: Replace "Arleqiun" with "Arlequin". Done

Line 187: Use "Defoliation analysis" as a third level heading instead of "ANOVA". Done

Line 205 and Table 3: Which is correct "LRIgel" or "LRIgiel"? I would suggest using a meaningful English abbreviation instead. Done

Line 206: Replace "Significant codes: 0 '***' 0.001 '**' 0.01 '*' 0.05 '.' 0.1 ' ' 1" with "Significance levels: ** – 0.01; *** – 0.001". Significants codes is corrected.

Line 208: Replace "similar" with "the same". Done

Lines 213 and 215: Use "allelic diversity" instead of "allelic stands". Done

Line 237 and Table 6: Please make sure whether the statement "bolded font above the diagonal" is correct. Also, the last column should be moved to the first position. Table 6 is corrected as recommended by the reviewer.

Table 7: Shannon-Wiener index (H') should be positive, not negative. True bug removed

Figure 3: It is a bit overloaded presentation of correlations: the (not necessary) colors are explained by the sidebar color scale, while the sizes of circles are not. I suggest removing colors and circles as a redundant information and making a simple plain text table instead. The layout of the figures and the colors used do not affect the substantive message. The size of the circles is proportional to the correlation scale, and the color used, as noted by the reviewer, is explained. After consultation, the authors ask for the current version of the figure. Similar figure used in publication https://www.researchgate.net/publication/265597288_An_investigation_of_Schema_theory_applied_to_the_biomechanics_of_the_sprint_start_in_athletics/figures?lo=1

Line 270: Replace "=" with "<". Done

Line 274: "range of defoliation is affected by tree genotype and phenotype" – By "range", do you mean the degree (extent) of defoliation? Also, I would doubt whether defoliation is affected by phenotype. Thanks for your suggestions for changes. Manuscript text corrected.

Lines 274-275: When referring to the "external factors affecting the plant" I would mention atmospheric pollution as one of the major factors affecting P. sylvestris. In the opinion of the authors, atmospheric pollution is included in the external stress factors (lines 334). The studied populations are located in a national park where air pollution is low (Rutkowski at. all 2019: Geochemical Referencing of Natural Forest Contamination in Poland, Forests, 11, 157 doi:10.3390/f11020157).

Line 281: "Stand age plays an important role in the health of stands in Poland" – Simply, it is just a common rule for biological species and not only in a given country. Text is corrected.

Lines 285-286: Replace "in which a well-developed a root system" with "with well-developed root systems". Done

Lines 286-287: "Trees growing in fertile soils, if exposed to water deficit, may be at significant physiological risk because of root system architecture". Here I would add that Scots pine in fertile soils is usually outcompeted by other forest tree species. Done

Line 289: "water deficit was observed at KNP" – Do you mean annual precipitation deficit? Then correct, respectively. Information on water deficit was supplemented in the text on lines 369-372.

Line 374: "Additionally, due to restrictions on the processes occurring in these stands" – What kind of restrictions do you mean here? If it is meant restrictions on human intervention, correct it, respectively .The sentence was corrected, it was made clear that there were legal restrictions in the intention of the authors.

Line 390: Replace "Table 1" with Table 1S". Done

Line 391: "Geographical coordinates phytosociological photos" – Do you mean "Geographical coordinates of phytosociological photos"? Done

Lines 392-405: Remove quotation-marks when providing your own statements. Done

References: Use boldface fonts for years of each reference. Done

Reviewer 3 Report

The manuscript “Comparison Biodiversity to Crown Damage in Natural Old Growth Pine Forests” compares the level of biodiversity, health and genetic diversity in old natural pine forests in Poland. The design here is quite original and provides support for a positive effect of genetic diversity on the response to stress. The analyses are mostly well conducted, but the main goal of the manuscript (is stand health affected by forest community diversity and genetic variation?) gets often lost in the text, which makes the manuscript difficult to read.  More focus on the main questions and some language editing will improve the quality of the text.

 The authors need to be cautious on interpretation based on the heterozygosities. They found a significant excess of homozygotes without discussing the potential reasons for this: presence of null alleles, genotyping errors, Wahlund effect, inbreeding… Further, the use of only 5 SSR loci is low to make conclusions about the role of heterozygosity levels on adaptation potential (section 4.3.2) and we are also missing the sampling size per population. It is indeed very surprising to see positive Fis values for old natural stands of a wind-pollinated species and the authors should first check (and eventually correct) for null alleles and potential genotyping errors before drawing any conclusion. Further, the correct definition for genetic parameters should be used: for example Fst provides an estimate of genetic differentiation, not a genetic distance. Referring to the literature for each parameter is necessary to know exactly which estimates and tests have been used and how the authors define it.

Below some specific comments:

L 38: “Genetic diversity of scotch pine…”

L 39 “ in pines…”

L40-41: this is indeed very likely, providing that the molecular markers are variable.

L43-45: difficult to understand, please rephrase

L45 Which selection? Natural selection or improvement of planted individuals?

L50 natural populations have only a lower risk of losing genetic diversity, providing that no strong selection pressure occurs and enough individuals are reproducing (no drift).

Figure 1: do you several stands in each research area? In this case how many stands?

L128: How many individuals were samples in each stand? This is very important to understand and evaluate the results

L134 “Molecular analyses were performed at 5 polymorphic microsatellite markers”

L174 Fst is not a genetic distance, but a fixation index providing an estimation of genetic differentiation. Please modify the text.

L187-184: this is not clear, are you conducting multiple linear regression or an ANOVA and why a linear model if you want to compare populations? Being a user of R, I think I know what you did and both models will provide anyway the same results. Maybe the focus of the analysis should be clear: are you interested on the significance of the population factor or on the effect of each population? (difference in the intercept)

Table 5: the Fis values are very high for a wind-pollinated species. Did you check for the presence of null alleles? Can you be sure that the tree sampled are not strongly related (minimum distance among trees)? This result should be more intensively discussed.

Table 6: you mean genetic differentiation? The significant values do not appear in bold

Author Response

We would like to thank the reviewer for the comments that allowed for a significant improvement in the quality of the article. The studied stands are among the oldest pine populations in Poland. Their persistence is also important, which favors natural selection processes increasing the value of homozygosity. A valuable note about the lack of null allele analysis has been corrected. Detailed bugs have been corrected as suggested, see below.

Below some specific comments:

L 38: “Genetic diversity of scotch pine…” Done

L 39 “ in pines…” Done

L40-41: this is indeed very likely, providing that the molecular markers are variable. The authors' intention was, in accordance with the cited literature, to indicate the variability of pine, and the molecular markers used in the research do not affect the variability of the species itself.

L43-45: difficult to understand, please rephrase In order to clearly understand the authors' intentions, the phrase has been corrected. The second part of the phrase specifies the tasks of forest management that select alleles.

L45 Which selection? Natural selection or improvement of planted individuals? As suggested by the reviewer, the phrase has been corrected. The text includes words indicating the selection carried out by forest management.

L50 natural populations have only a lower risk of losing genetic diversity, providing that no strong selection pressure occurs and enough individuals are reproducing (no drift). The sentence has been corrected as suggested by the reviewer. Information is provided on possible causes of selection in natural populations.

Figure 1: do you several stands in each research area? In this case how many stands? L128: How many individuals were samples in each stand? This is very important to understand and evaluate the results

The value of 50 trees in each population from which plant material was collected for genetic analysis was specified and introduced.

L134 “Molecular analyses were performed at 5 polymorphic microsatellite markers”

The sentence has been corrected as suggested by the reviewer.

L174 Fst is not a genetic distance, but a fixation index providing an estimation of genetic differentiation. Please modify the text.

The sentence has been corrected as suggested by the reviewer.

L187-184: this is not clear, are you conducting multiple linear regression or an ANOVA and why a linear model if you want to compare populations? Being a user of R, I think I know what you did and both models will provide anyway the same results. Maybe the focus of the analysis should be clear: are you interested on the significance of the population factor or on the effect of each population? (difference in the intercept). Thanks for the remark, this part in the methodology has been corrected. We used an ANOVA model to compare population means. Example r code: summary(res_aov <- aov(Def ~ Population, mydata)).

Table 5: the Fis values are very high for a wind-pollinated species. Did you check for the presence of null alleles? Can you be sure that the tree sampled are not strongly related (minimum distance among trees)? This result should be more intensively discussed.Thank you for paying attention to the high value of Fis in the analyzed populations. They may be a consequence of the presence of null alleles, which we confirmed. We supplemented the information about them by adding for supplementary number 3 and supplementing the information in the text with lines 274-277.

Table 6: you mean genetic differentiation? The significant values do not appear in bold

The sentence has been corrected as suggested by the reviewer.

Round 2

Reviewer 3 Report

I have reviewed the manuscript “Relationships between some biodiversity indicators and crown damage of Pinus sylvestris in natural old growth pine forests”. The authors answered most of my questions and provided further required information. Although they provide the explanation for the significant excess of homozygotes (presence of null alleles), the authors did not modify the discussion accordingly (section 4.3.1). The low heterozygosity is artificial here because almost all loci have null alleles, making impossible to conclude for the presence of inbreeding or whatever effect of the mating pattern. One option could be to correct for the presence of null alleles (Van Oosterhout, Cock & Hutchinson, W. & Wills, Derek & Shipley, Peter. (2004). Molecular Ecology Notes. 4. 535-538). The text is still difficult to follow, needs polishing and there seem to be problems in the editing software because the text in red sometimes makes no sense (see L45-46, L50-51…).

Author Response

The authors of the publication think, like the reviewer, that one of the reasons for the low heterozygosity of the studied populations is the presence of null alleles. These alleles do not allow the recognition of heterozygotes containing such an allele (Callen et al. 1993). Although the statistical methods used to detect null alleles may be biased and give false positives results (Dabrowski et al. 2013). Null alleles are common in microsatellite markers. For example Scotti et al. (2005) described them for 0.2 of the analyzed loci. The frequency of null alleles in the studies described above is similar to that in the present article. Also other research carried out on the seed orchard in Poland has a similar null allele value (Przybylski 2016). In order to detect null alleles, additional studies would be needed to segregate alleles in endosperms, while the data correction suggested by the reviewer would be the reason for the reinterpretation of the all text.

On the other hand, null alleles are not the only reason for the increased homozygosity of the studied populations. The authors think that the homozygosity obtained results mainly from the age of the forest stands and their natural origin, which was supplemented in the text in lines 441-450. This context is important for the interpretation of the main goal of the work, which is the analysis of the relationship between the biodiversity of the studied stands and the health condition of tree crowns. The analysis of genetic variability of stands is the added value of the article.

Reviewer notes on the logical consistency of sentences are included on lines 45-51.

References:

Callen D.F., Thompson A.D., Shen Yang, Phillips H.A., Richards R.I., Mulley J.C., Sutherland G.R. (1993). Incidence and origin of "null" alleles in the (AC)n microsatellite markers. Am. J. Hum. Genet. 52: 922-927.

Dąbrowski N. J., M. Pilot M., M. Kruczyk M., M. Żmihorski M., Umer H. M., Gliwicz J. (2013). Reliability assessment of null allele detection: inconsistencies between and within 2 different methods. Mol. Ecol. Resour. Mar,14(2):361-73. doi: 10.1111/1755-0998.12177.

Przybyski P. (2016). Zmienność genetyczna wybranych plantacji nasiennych sosny zwyczajnej (Pinus sylvestris L.) w aspekcie błędów przypisania szczepów do drzew matecznych. PHD manuscript, manuscript IBL library.

Scotti I., Burelli A., Cattonaro F., Chagné D., Fuler J., Hedley P.E., Jansson G., Lalanne C., Madur D., Neale D., Plomion C., Powell W., Troggio M., Morgante M. (2005). Analysis of the distribution of marker classes in a genetic linkage map: a case study in Norway spruce (Picea abies Karst). Tree Genet. Genomes 1: 93-102.

This manuscript is a resubmission of an earlier submission. The following is a list of the peer review reports and author responses from that submission.

Round 1

Reviewer 1 Report

The manuscript entitled “Impact of biodiversity on crown damage in natural old growth pine forests” tries to correlate biodiversity indicators with genetic diversity and defoliation of Pinus sylverstris unmanaged forest stands.

Although the idea is quite good, the manuscript needs severe correction in the central theory of the manuscript. When you conduct a genetic diversity analysis in an unmanaged forest population you must discuss first of all forces of natural selection that shaped it (gene flow, mitigation, bottleneck effect etc). Furthermore, whether authors present age of Pinus sylverstris, they did not provide the method from which the calculation was made (did they collect drills?). From my point of view, they do not also discuss other factors and historical data about health of forest stands, they could also provide meteorological data that could be associated with genetic and biodiversity indicators. Additionally, they need to improve the discussion with other references that present genetic diversity of forest trees, adaptation and biodiversity.

Below find some minor and major considerations:

Line 20: delete “there”

Line 22: rewrite he whole sentence

Line 23: delete “in which they grow”

Line 35-37:  Is there a reference which provides this separation, to me it is really unknown and we do not use such a separation “Differences between populations at the gene level are considered as genetic diversity, while intra-population diversity is defined as genetic variability.”

Line 43: correct “characterizes” with “characterized”

Line 45: “The maintenance of biodiversity in afforested areas reduces the risk of inbreeding, which can reduce the natural regenerative capacity of individual species”. It does not make sense the part that natural regenerative capacity is reduced. please rewrite it.

Line 55: corrected “lowered” with “is low”

Line 60: replace “it may lower” with “genetic variability presents lower values”

Line 72: replace “analyzes” with “analysis”

Line 87: after “used” add “in order to”

Line 98: erase “which the observed populations are growing”

Line 229: correct title “biodiversity” with “diversity” and the same in Table 6

Line 246: erase “The carried out” and replace “groups” with “grouped”

Line 259: the table is not well designed and also statistically significant are now well visible, please indicate with an asterisk.

It does not make sense such a comparison to me; when you compare number of species and number of private alleles, you conducted the analysis only for Pinus so if you have low number of private alleles this is not an indicator for tree, population of for ecosystem health.

Line 332: replace “remembered” with “noted”

Line 340: in which root system is already??

Line 353: replace “while no such undertaking” with “while not such a list”

Line 383: please rewrite the sentence “with genes subject to negative”

Line 387: allele frequency? You did not provide any table or figure for allele frequencies.

Line 401: replace “is insignificant” with “was not significantly different and was below 1%”

Line 420: replace “in adapting” with “in adaptation

Line 419: epigenetics also can have a severe impact on adaptation and also there is evidence in literature that phenotypic plasticity is a result of epigenetics. You do not mention tis factor in your manuscript.

Reviewer 2 Report

The authors studied the effect of biodiversity on phenotypic responses to local environmental conditions (e.g. crown defoliation) in two national parks in Poland. The manuscript focusses on the analysis of very old trees (with an average age of pine trees of over 100 years), which raises the question on how much the phenotypic variation seen is actually caused by age or local adaptation rather than genetic differences. To me, the argumentation in the introduction and discussion and the results of the study could not sufficiently convince me that such phenotypic plasticity should be described based on genomic differences rather than e.g. transcriptional differences (due to adaptation or age-dependent immune deficiencies) or at least correlations with quantifications of local conditions (e.g. over a certain period). The introduction and discussion feel very lengthy. It should be considered to use subtitles in the discussion that summarize the obtained result and put it in perspective. In the current version, it is very hard to filter out the key messages of the manuscript. The same holds true for the table descriptions, which are very short and do not bring the data into context.

Some minor comments:

  • Line 65: the sentence does grammatically not sound correct
  • The project hypotheses are too broad in my opinion and scientifically not sound. For example, what are the "numerous similarities in ecological growth conditions and selection processes" you are refering to and why should there be no genetic diversity behind it, keeping in mind that immunity-related genes (e.g. R-genes) are under strong selection pressure.
  • Line 146: the company name needs to be changed into Macherey-Nagel
  • Line 150: The sentence does not sound complete
  • Line 154/155: Correct for degree and C
  • The PCoA listed in the Annex were not available 
  • Fig. 4: How can there be no standard errors if the data comes from average of plants?
  • Fig. 6: Please specify the method used for the correlation analysis (e.g. Pearson). 
  • Line 340: the statement is not correct- root systems are always developed otherwise plants would not get any nutrients or water and die. 
  • Line 427: This statement is not put into context. Please refer to the table or figure in your manuscript you are rooting this statement on. Furtheremore, I think that results from trees with an average age of over 100 years can not be used to make statements related to a "relatively short time span".